# Healthier Construction: Conceptualising Transformation of Mental Health Outcomes through an Integrated Supply Chain Model

Kerry Anne London [1,*][ID], Tanya Meade [2] and Craig McLachlan [1]

[1] Office of Research and Innovation, Torrens University Australia, Adelaide 5000, Australia; craig.mclachlan@torrens.edu.au
[2] School of Psychology, Western Sydney University, Sydney 2150, Australia; t.meade@westernsydney.edu.au
* Correspondence: kerry.london@torrens.edu.au

**Abstract:** The construction industry is undoubtedly one of the most significant global sectors that contributes to sustainable development across physical, social, environmental and economic objectives. Globally the value of the construction industry is USD 10 trillion annually. The robustness of the sector is in serious question with a crisis in mental health. The rebuilding of economies is often led by significant capital works programs and therefore in response to the global pandemic, it is anticipated that this problem will only be exacerbated. The construction sector has a unique project-based structure of numerous intersecting subsectors, which influence the behaviours and culminate in highly demanding work environments on a project-by-project basis. We propose that to institute transformational change to the mental health problem, we need to challenge current problematisations towards presenting a new conceptual framework. The aim of this paper is to analyse the industrial organisation and the structural and behavioural context of the industry and propose a new approach to understanding interactions at multiple levels in relation to root causes of the mental health problem. Aligned to the UN SDG that we are to ensure healthy lives and promote well-being for all, this paper responds to high rates of depression, anxiety and suicide in the construction industry. There is a need to generate new knowledge about the interactions between multi project supply chain, construction project supply chain environment and construction supply chain performance in relation to mental health outcomes. Literature indicates that there is a wealth of research on stressors, coping and interventions at an individual level, however very little from an 'insider' construction management perspective which contextualise mental health outcomes with the environmental stressors. Coupled with this, past research designs predominantly utilised quantitative approaches reliant on questionnaires. We critique past problematisations of the mental health problem and show how it has been represented to enable the development of a reframed conceptualisation. There is a need to identify contextual evidence-based stressors throughout the construction project supply chain. We present a transformational change model integrating construction industry specific context knowledge with psychosocial expertise to improve workers' mental health. Future research could lead to outcomes including recommendations and guidelines to engage management actors who can influence positive change through preventative strategies leading to effective and measurable mental health and project performance improvements.

**Keywords:** mental health; depression; anxiety; suicide; construction industry; industrial organisation; environmental stressors

## 1. Introduction

The Sustainable Development Goals (SDGs) were adopted at the United Nations Conference on Sustainable Development in Rio de Janeiro in 2012 in order to focus international efforts on addressing urgent environmental, political and economic challenges facing the world [1]. These UN SDGs are an expanded comprehensive framework and include 17

Goals that are importantly interconnected, meaning that success in one goal will have a flow-on effect on other goals [1–3]. For example, dealing with the threat of climate change impacts how we manage our fragile natural resources, achieving gender equality or better health helps eradicate poverty, and fostering peace and inclusive societies will reduce inequalities and help economies prosper [2–6]. One of the SDGs (no. 3) is to "Ensure healthy lives and promote well-being for all at all ages" [7]. Mental health and psychosocial wellbeing were defined by the WHO in 1978 as an integral part of health and have been addressed in many UN resolutions. However, it is only recently that mental health has been included as a part of the unified global agenda starting with the world leaders adopting the Sustainable Development Goals (SDGs) in 2015 [7]. Part of that commitment was to prioritise "prevention and treatment of non-communicable diseases, including behavioural development and neurological disorders, which constitute a major challenge to sustainable development" [7,8].

It is estimated from various studies that the annual global mortality rate, from suicide ranges between 758,000 to 884,000 with suicide being the third leading cause of death among young people [9,10]. Further, it is estimated that one in five working-age adults has a mental health disorder [11,12]. There is however a growing recognition within the international community that mental health remains one of the most neglected yet essential development issues in achieving internationally agreed UN SDGs [13].

As most people with mental health problems are employed, there is an economic cost that will be borne by the organisation through reduced productivity [14]. Productivity can be managed through addressing workplace contextual factors that cause job stress and perceived lack of job security and control [15,16]. One of the largest world economies is the construction sector with about USD 10 trillion spent on construction-related goods and services every year [17]. Physical infrastructure development provides a critical economic lever that governments draw on in times of need and in response to economic crises, the most recent one being COVID-19. The construction sector is expected be worth USD 15 trillion dollars in 2030. As such, it employs a significant workforce and therefore can play a significant role in contributing to the SDGs and particularly the health and wellbeing of its workers [18,19].

In Australia, the construction industry directly employs 8.6% of the country's total workforce [20]. Construction workers are more than twice as likely to attempt suicide as other Australians, six times more likely to die by suicide than through a workplace accident and 21% have experienced mental health problems [21,22]. Apart from individual and family costs to construction workers; there are economic costs of mental health problems to the construction industry with estimated Australian annual cost of almost AUD 11 billion dollars [22].

Leaders of the Australian construction industry, similar to other countries, have sought to identify and address the mental health issues and suicide in its workforce. Subsequently, the Australian Building and Construction Industry Blueprint 2018–2022 [23] was developed from a roundtable of industry and government stakeholders to provide a framework to inform understanding of the challenges faced by the sector and undertake focused and timely actions to address those challenges. Specifically, an undertaking was made that the construction industry: (1) would benefit from understanding the challenges facing its workers, (2) should target the areas where mental health improvement could be efficiently and cost effectively achieved, (3) should learn from other industries, and (4) should display leadership, and work to an agreed roadmap and timetable.

The following were part of the shared vision [23] to be achieved by 2021: (1) the industry was a global leader in mentally healthy workplaces, (2) had a nationally agreed framework that was adopted and supported by all stakeholders, (3) has accepted mental health as a normal topic of conversation on site, (4) receives reports from organisations on key metrics, and (5) supports and participates in a growing evidence and research base around suicide and mental health. The industry is certainly not a global leader in mentally healthy workplaces, and it is questionable if the other objectives were achieved. There is

little evidence to suggest that there has been an evaluation to establish the extent to which the shared vision was achieved.

Further, the group also developed a six point plan which included the following: (1) to put an effective national work group and communication strategy in place, (2) complete research to define a mentally healthy workplace and identify risks for mental health and suicide in the industry, (3) develop and publish an industry mental health intervention blueprint, (4) agree guidelines with regulators with a view to making future legislative change, (5) develop a strategy for full industry engagement over the next three years, and (6) to ensure that all apprentices under structured training receive mental health and wellbeing education.

Various systemic construction industry factors were foregrounded as being critical to affecting workers' mental health and suicide risk, including compressed delivery demands on projects, the complexity of competing agendas, lack of consistent government leadership and an industry structure that often focuses on short-term solutions. However, the proposed plan focussed on mental health intervention programs but did not include consideration of the industry systemic causes.

In late 2021, with a similar agenda, another industry wide consultation was conducted on a Culture Standard [24] and the following reforms were recommended:

Wellbeing–prioritise the mental and occupational health of the workforce through programs such as resilience training and suicide prevention, providing 'mental health first aiders' on site and creating program schedules that prioritise worker wellbeing.

Diversity and Inclusion–attract and retain a diverse range of people to work in the industry by addressing pay gaps, involving women in strategic decision-making roles, providing suitable amenities, and removing offensive material in the workplace.

Time for Life–ensure workers have enough time to rest and pursue activities outside work through initiatives such as flexible working arrangements and project scheduling that ensures workers are only working five days per week (or no more than 55 h per week) and wherever possible, not on weekends, as is presently the case.

Interestingly, the outcome of this consultation was similar to the previous consultation with the recommended reform being solely aligned to employee mental health management (once it is identified) to the exclusion of critical environmental factors. That is, a deeper examination of the underlying structural and behavioural conditions of the industry is necessary to identify factors that give rise to pressures at the individual project level and create inherent stresses to the value chain. Those cannot be readily resolved by individualistic, isolated ad hoc approaches which in the absence of understanding the context and its role in the occurrence of mental health issues in the workplace may fall short of sustainable solutions and indeed exacerbate the problem.

Numerous multi sectoral mental health reforms have been undertaken and resulted in various landmark reviews, including the Productivity Commission Inquiry into Mental Health, the Royal Commission into Victoria's Mental Health System, and the National Suicide Prevention Adviser and Taskforce. These have collectively identified the scope of the mental health issues, research informed understanding of the nature of those mental health challenges and the impact on the individual and the society if those are not addressed, and a range of prevention and intervention programs that have been implemented across the sector. Further, those reports show demonstrable and ongoing commitment to finding effective solutions to mental health issues in the construction industry.

The aim of this paper is to explore construction industry workers' mental health in the context of individual and immediate project work environment factors within the context of industry structural and behavioural characteristics and to provide a conceptual model to inform and transform change in the immediate workplace environment and across the sector as a mental health preventative strategy. In doing so, we are proposing a move away from focussing on addressing the mental health problem with individual counselling and education programs in isolation from the work context structures. We propose that the fundamental sectoral industrial organisation and underlying structures may influence the

behaviours in the sector thus at time may contribute significantly to the stress, distress and mental health problems in its workforce which is predominantly one of the most vulnerable population groups–young, low-skilled men. The methodological approach is twofold. First, it is to review the statistics related to mental health outcomes in the construction industry, and the academic literature on the workplace stressors in the construction industry. Second, it is to map the mental health risk points to the construction industry project-specific timelines with the corresponding stress points, noting and exploring the interplay between the project and the individual stress points experience. Subsequently by creating a model of interplay between a project and a person, we can map the relevant literature and inform future research to identify how the mental health issues in the construction industry can be understood and addressed at a more comprehensive and contextualised levels.

## 2. Literature Review Critique

Many solutions to mental health challenges in the construction industry are in the form of mental health intervention programs designed to provide counselling and assist individuals. We argue that while there is much value in this approach, there is a need to consider other models that involve a multi-level conceptualisation. We now explore the literature to examine the extent of the literature that approaches the problem from this perspective and addresses the inter relationships between individual, workplace and industry.

### 2.1. Mental Health

Mental health problems are a significant concern in any workplace and are associated with personal and productivity losses, and work safety [25]. The prevalence of mental health problems in construction industry is considerably higher compared to other industries [26]. In addition to reported anxiety, depression and substance use disorders, there is evidence of high risk of suicide with construction workers being six times more likely to die from suicide than an accident at work [27,28]. Those suicide rates remain one of the highest across industries with one death by suicide occurring every second day or 24.6 per 100,000 compared to 13.4 per 100,000 in non-construction industry [29–31].

There is a higher rate of suicide among construction workers when compared to non-construction workforce. Most at risk are younger, low-skilled construction workers [32]. Further, construction workers in general report higher levels of alcohol abuse (17%), illegal drugs use (14%) and suboptimal sleep (9%) which are associated with poor mental health [33]. Similar findings were reported in an earlier systematic review [34] which grouped identified risk factors for anxiety and depression into individual factors (poor health and lifestyles), team environment (unsupportive workplace relationships) work conditions (job overload and job demands) and work-home interference (work pressures impacting on home life).

Psychological distress in construction workers is reported at higher rates, compared to the national estimates, particularly so for men aged 25–44 who are less confident than older age groups (over 44 years of age), in knowing how to seek help or how to speak to their colleagues about their psychological distress [35]. At the individual level are possible barriers to self-acknowledging psychological distress and health seeking, possibly due to perceived stigma, masculine culture and job insecurity [33].

While younger construction workers have lower suicide literacy and mental health understanding, they (particularly the 15–24-year-olds but also 25–34-year-olds) are more open to change and to endorse the workplace's role in preventing and assisting in mental health, compared to older age groups or professional construction workers group which suggests there are age and work role differences [32]. These differences need to be considered in development of workplace-based prevention and interventions strategies.

*2.2. Workplace Stressors*

Workplace stressors and the coping strategies associated with psychological distress include lack of personal time and long working hours and weeks, which are indicative of a poor work–life balance [36]. There are other reported individual and workplace factors that may be associated with psychosocial hazards. Psychosocial hazards are defined as work related stress, which can lead to psychological or physical harm. Such factors include high-pressure work, job insecurity, and project cost and time completion schedules [25,36].

Two recent systematic reviews identified numerous psychosocial hazards in the construction industry. Chan et al. [37] identified and extracted 32 (from 16 pooled studies) mental health risk hazards. Of these the three most reported hazards were workload, job control and family-related concerns followed by income and job security and poor coping skills as also being associated with mental health problems.

While considerable research has been conducted on the work stressors, less is known about the psychosocial factors such as marital status, relational frictions, loneliness and bereavement [37]. Further, psychological conditions such as PTSD are also more prevalent in the construction industry. Typically, PTSD is the result of a previous work incident that could be defines as a hazard or with non-work-related life experiences [37]. The authors suggest that it is important to also identify protective factors as means of actively promoting better health [37].

In another systematic review, Tijani et al. [38] identified 49 hazards associated with mental health problems (from 38 pooled studies). Across those studies they identified: work overload (25/37 studies), home–work (inter-role) conflict (21/37 studies), poor working environment (17/37 studies) and poor working relationship (16 out of 38 studies) as the four most common factors. The authors proposed a conceptual framework of organizational and environmental stressors that lead to occupational stress and include organizational and physical stressors (18 out of 49 hazards identified), task stressors (11 out of 49 hazards identified), personal stressors (12 out of 49 hazards identified) and gender-related stressors (3 out of 49 hazards identified).

Sun et al. [25] in a systematic and meta-analysis review of 48 studies across Asia, North America, Africa, Europe and Oceania has ranked psychosocial hazards frequently associated with mental health problems focusing on construction industry and including unskilled, trade and professional workers. Their review identified 14 psychosocial hazards (eight job demands and six job resources) of which all, but one was found to be associated with mental health problems; the top four hazards being role conflict, role ambiguity, job insecurity and interpersonal conflict with some noted differences across countries and the trade and professional workers. The authors reported that the job demands hazards were associated with worse mental health outcomes compared to the job resources hazards, which suggests that reduction in excessive work burden if addressed could prevent or minimise mental health problems in the construction workers.

In addition to mental health issues, construction work is also physically demanding and therefore associated with physical injury, work-related ill-health and disability, accounting for 12% of all fatalities in the workplace. Furthermore, this accounts for 9% of the workforce; and for 37% of compensation claims due to physical injury, as well as high rates of burnout and early retirement [27]. Further, the work-related physical pain is found to be associated with diminishing mental health and work ability [39]. Concerningly, those workers who have poor self-rated health due to the demands of their work, and thus have a pool locus on internal control [27]. Pain can lead to additional pressures and stressors which in turn can be experienced as psychological demands and subsequent distress. Turner and Lingard [27] suggest that an integrated approach to occupational health and safety with a focus on both physical and psychological wellbeing, to be addressed holistically and concurrently.

As a result of the studies reviewed, there is evidence to suggest that there is a need for a more discerning approach to understanding the interaction between individual, the immediate project work environment factors, the broader underlying structural and

behavioural industry context, and mental health outcomes. The key limitation of past research in construction mental health is the focus on generic approaches to mental health issues with a cursory understanding of the work environment stressors and industry enablers and barriers for change [39]. In summary to address workplace stressors and support better physical and mental health of the workers, it is necessary to understand the construction industry context and potential enablers and barriers to change that may be needed to better support its workers' wellbeing.

### 2.3. Project and Industry and Project Context

While there is considerable research literature on workplace stressors, coping and interventions at an individual level, limited consideration has been given to the management perspective and construction project supply chain context. Tijani et al. [40] reviewed 60 studies on mental health research in relation to construction project professionals and identified several research gaps. Namely, while construction project professionals experience high levels of depression anxiety and substance abuse, like unskilled construction workers, generic rather than project, role and responsibility specific stressors were considered as potential root causes of those mental health issues. The authors suggest that further research should focus on project-related stressors to identify potentially harmful project management practices and subsequently develop 'mentally healthy project organisational designs' for construction project professionals.

The findings of this review [40] further highlight the need to move to more specific interrogation of the workplace context in which workers across different roles experience mental health issues and to also be specific about the range and nature of mental health issues experienced by the workers and the potential impact of those issues on project management and performance.

Further it is important to delineate the construction industry contextual differences relating to the nature of work and roles, the size and the location of the organizations. For example, in a study of 1124 mining and construction workers in remote regions, the most reported stressors were missing special events (86%), relationship problems (68%), financial stress (62%), shift rosters (62%) and social isolation (60%) which were significantly related to high psychological distress [41] and subsequently need to inform tailored prevention or intervention programs to the industrial context and workers' needs.

## 3. Conceptual Model and Discussion

To respond to high rates of depression, anxiety and suicide in the construction industry, there is a need to generate new knowledge about the interactions between (a) industry structure and behaviour (b) project environmental characteristics and (c) project performance metrics in relation to mental health outcomes and how the individual is embedded within project, organisational and industry supply chain context (refer to Figure 1).

We present a transformational change model integrating construction industry specific context knowledge with individual psychosocial context to improve workers' mental health. Future research could lead to outcomes including recommendations and guidelines to engage management actors who can influence positive change through preventative strategies leading to effective and measurable mental health and project performance improvements.

The construction of any built system is a complex problem involving numerous firms who temporarily work together in response to individual projects–or so it seems [41]. For the specific construction related activities this has led to the concept of subcontracting. It is well accepted in the property and construction industry as well as in academia that projects involve the practice of continual association and disassociation, that is, forming and reforming for individual projects. This specialisation, subcontracting and bespoke project orientation is indeed the common mode of economic organisation within the property and construction industry for construction trade subcontracting and the vast design, planning and management consultancies. Many of the functional and technical specialisations required for project contracts reside within numerous individual firms which are coordinated

on a project-by-project basis. Firm interdependencies can become particularly acute since construction industry firms rarely act as isolated and independent entities. The degree to which coordination is required may vary from project to project and from country to country, as the degree of specialisation and vertical integration can differ across locations and projects [42].

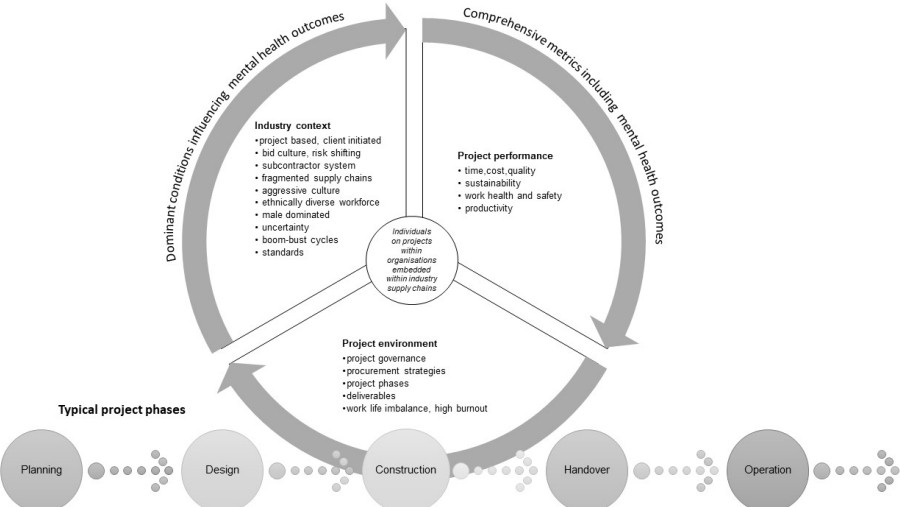

**Figure 1.** Conceptual Transformational Model for Better Health Outcomes for Construction.

There are important considerations for organisational, project and workforce management that this project-based mode of organisation foregrounds. The industry is not entirely atomistic as there are so many variations to the timeframe of relationships, i.e., some of the relationships are not simply a 'one-off' individual project contract but are embedded in both (a) a series of multiple projects simultaneously and (b) a series of multiple projects over a lengthy period of time. The links between organisations are tied to project contracts and hence the conditions of the individual project influence the environment of the project. However, the importance of the services and/or products provided by firms and the countervailing power relationship between organisations presents both opportunities and challenges in enabling change between the organisations and thus across all construction supply chains.

The complexity and multiplicity of the web of procurement relationships at all levels of the construction supply chains is influenced by key interacting attributes: formation based, transaction significance and then negotiation strategies based upon risk, expenditure, level of control and countervailing power [42]. Procurement strategies and tendering are important levers to behavioural and structural change. Key to transforming the industry to achieve transformation change for mental health outcomes is an awareness and understanding of these levers. Particularly important is understanding how the levers can impact performance metrics on projects. The tendering process is key to influencing the pursuit of individual project contracts and then delivery and monitoring of that contract however the specific project and the way in which it is embedded within the context of other medium and longer-term relationships between organisations and then the anticipation of other future contractual relationships [42]. Thus, the fundamental act of procurement is an important building block to understanding when, where and how transformational change in the industry can take place. Regulation of course is another key lever that one can interact with to reduce stress across the system. We believe the more enduring and sustainable change on the state of an industry is achieved by changing the way in which actors within the system behave; rather than using the punitive threat of regulation which in itself is resource intensive to activate, develop and continually monitor. The merits of one versus the other is a worthy debate.

The construction industry is characterised by certain drivers working at an industry and project level. It is critical to understand the inter relationship between drivers, as well

as the interrelationships between the time at text, which the greatest impact can impact positive change to reduce system stresses. The ability to have measures in place that allow one to track and adjust systems is very important. Instinctively many long-term experienced practitioners both in public and private sector have intuitions about the 'feel' of the industry but there is such a paucity of evaluated and tested measures. There is a need for data and aligned directions for policy and practice change. Industry proponents continue to acknowledge the challenges and yet feel powerless to effect change. To effect change it will require a transformational collaborative effort. At the end of the day the industry and all the firms will require evidence, tools and guidance to effect such large-scale change.

Table 1 proposes an interweaving of the four dimensions of industry context, catalysts, future state and performance across three domains of environments which have their own unique drivers or barriers to change including multi project supply chain environment, construction project supply chain organizational environment and construction supply chain project procurement environment. The table presents the construction industry specific factors that may affect or exuberate mental health outcomes for its workers. The four dimensions identify the inter relationships between factors describing the dominant structural and behavioural characteristics of a project-based sector, the dominant timeframe and actions effectively impact systemic change, directions for policy and practice influencers and finally initial and future state measures of likely systemic failure. Each of the four dimensions are then explored cross three specific bounded knowledge domains that either inhibit or drive change to mental health outcomes including an industry level, a project organisational level which incorporates intercultural and organizational environments, and finally project procurement level.

This approach builds upon and expands the past work with identified stressors and deepens the discussion on root causes. The mental health program intervention idea is taken further upstream. It is not enough to provide individual mental health education and counselling programs to intervene at an individual level; an intervention program is required to address and intervene in the workplace environment and provide leaders, executives and managers a decision framework and guideline to fundamentally change the direction of entire projects and portfolios of projects at an organisational level.

Individuals at an operational level can be powerless to effect change and/or powerless to have their voice heard. Even individuals leading projects often lack control, particularly if there are more systematic controls and institutional mechanisms that are integrated within a project. Such controls ideally would be comprehensive and predict mental health risk over the life of a particular project. People in the construction industry can work on more than one project at any one time and thus they are often embedded with quite different groups of work colleagues both within their own firm and external to their organisation. Different projects have different timelines and pressures and, in many cases, although one may deliver on milestones at a particular point in time on one project there is often little or no downtime till another deadline emerges, or when stress points arise from another project. The cumulative pressure can be intense and relenting.

There is a real need to develop new methods for modelling and prioritisation of resources that are less stress intensive on the workforce. In our model we have identified key areas for reform including one dimension of performance measurement. There have been numerous discussions in literature on supply chain performance. The most recent analysis of the literature in 2018 (Reddy et al.) of over 215 articles from 1998–2018 comprehensively indicates that within the supply chain performance measurement topic there has been a dearth of studies focussed on simulations. The authors observed that performance measurement in the context of supply chains is still a fruitful area to carry out future research. The authors key finding was that additional research is required in the area of supply chain performance modeling using simulation techniques like system dynamics and discrete event simulation useful for supply chains operating in a volatile environment. Significantly the paper was silent on different industrial organizational characteristics that various sectors have such as project-based vs. long run manufacturing.

Table 1. Conceptual Definition of Multilevel Multi-project Systemic Transformational Model for Better Health Outcomes.

| INDUSTRY CONTEXT: Dominant Structural and Behavioural Characteristics of a Project-Based Sector | CATALYSTS: Dominant Timeframe and Actions Effectively Impact Systemic Change | FUTURE STATE: Directions for Policy and Practice Influencers | PERFORMANCE: Initial and Future State Measures of Likely Systemic Failure |
|---|---|---|---|
| **Multi Project Supply Chain Environment** | | | |
| High workloads | Planning Phase: Initial starting conditions of projects for each firm | Development of tools to monitor Development of benchmarking databases | Project and organisational capacity to deliver Monitoring of project multiplicity |
| Industry structural conditions of project bid and work uncertainty exacerbating job insecurity | Pre-planning Phase: Annual development of pipeline presented to industry by major private and public sector clients | National database of projects mapped to mental health metrics (absences, productivity, hours worked) | Volume of projects across a sector in a region |
| Uncertain economic environment resulting in poor decision making on project true cost resulting in multiple projects that are economically unviable exerting pressure on firms' staff & supply chains | Pre-construction and tendering phase | Analysis of case studies of low bids and poor mental health outcomes | Establish comparison between poor health outcomes through metrics and relate to costs projected in tenders |
| Uncertain economic environment may lead to bankruptcies resulting in redundancies | Tendering: Bid analysis by clients. Project completion: post construction & handover cross project assessments and performance evaluations of key supply chain actors | Analysis of case studies of bankrupt companies and impact on poor mental health outcomes | At the firm level significant numbers of project submissions with highly significant reduced fee/tender value in comparison to market bidders |
| Emerging significant infrastructure workload as a peak arising from COVID pandemic as governments seek to re-energise the economy | Pre-planning Phase: Annual development of pipeline presented to industry by major private and public sector clients | Inter-state comparison of workload and workforce capacity | State by state analysis of workforce capacity |
| **Construction Project Supply Chain Organisational Environment Level** | | | |
| Project specific demands with peaks and troughs of pressure points within strict budget and quality regimes | All project phases | Recommendations on project portfolio planning and management and scheduling supported by case study evidence | Examples of system capacity |
| Project specific contracts with high performance metrics of timely completions | Planning and Tendering | Guidance on various national and international approaches to limits and ranges to project timeframes | Metrics on limits and ranges to project timeframes within Australian context |
| Cascading pressure down fragmented subcontract contractual supply chain | Tendering, Interim Milestones Handover and Completion | Project by project benchmarking on construction supply chain workforce cohesiveness and capacity to deliver on project | Integration supply chain metric |

**Table 1.** *Cont.*

| INDUSTRY CONTEXT:<br>Dominant Structural and Behavioural Characteristics of a Project-Based Sector | CATALYSTS:<br>Dominant Timeframe and Actions Effectively Impact Systemic Change | FUTURE STATE:<br>Directions for Policy and Practice Influencers | PERFORMANCE:<br>Initial and Future State Measures of Likely Systemic Failure |
|---|---|---|---|
| Organisations balancing multiple projects simultaneously with competing work deadlines | New bids (design and construction) and periodic monitoring | Recommendations on project portfolio planning and management and scheduling supported by case study evidence | Examples of system capacity |
| Government as a significant influencer as a client through procurement of major public sector infrastructure projects | Design and Construction pre-bid and tendering | Case studies of best practice in relation to procurement directions aligned to mental health outcomes across entire client portfolio of capital works | Inter-state metrics and benchmarking on public sector performance in relation to mental health outcomes related back to procurement methods |
| **Construction Supply Chain Procurement Environment** | | | |
| 'Silence' in tendering criteria of psychological work health and safety outcomes | Design and Construction pre- bid and tendering | Best practice examples of procurement and tendering directions aligned to mental health outcomes | Examples of project tendering metrics |
| Frequent changing work specifications during all phases of the project to meet uncertain and complex project conditions | Design and Construction reporting and monitoring | Guidelines on impact of decision-making flow on effect including quantitative and qualitative evidence | Metrics on change work orders and stress impact on workforce to deliver |
| Lack of leadership with incentives for workplace well-being and a 'complete the job' at all cost mentality | All phases but particularly project delivery milestones | Guidance for senior/middle management supervisors of project directors/managers within consultancies/contracting companies | Leadership performance indicators inclusive of project and staff mental health outcomes |
| Extensive long work hours during the work week with little downtime | Design and construction phases exacerbated at organisational milestone delivery submission times | Case studies of projects with varying work hours aligned to productivity analysis | Recommendations for companies and clients on feasibility of reduced working week and associated impacts on completion time and costs |

The authors however did recommend some generic steps for decisions makers to assist with identifying the most suitable supply chain performance management system and the right performance measures for their organisations including: (a) Identify the company supply chain strategy and objectives; (b) Identify the right performance measures and performance management system based on the supply chain strategy and objectives; (c) Prioritize the selected measures with the focal supply chain strategy; (d) Inter-relate the key performance measures with the supply chain strategy subsequent to discussions with the stakeholders; and (e) Develop a suitable supply chain performance management system and explain to the other members in the supply chain to evaluate the performance management system.

Although there were descriptions of selected simulation-based research models and a strong advocacy for more simulation-based research, the paper while providing a comprehensive review, did not offer conceptual or theoretical guidance for future research.

## 4. Conclusions

In conclusion our literature review indicates that there is a significant pool of research already on individual worker stressors, coping and interventions in the construction industry. Importantly through our literature mapping, we have identified gaps from an 'insider' construction management perspective related to project management as opposed to the individual worker. An insider construction management can significantly affect the mental health and wellness of an employee. Therefore, it is possible to contextualise mental health outcomes with respect to environmental stressors across construction projects. We have explored the complexity of factors that can contribute to worker behavioral stress and mental health responses across construction industry procurement. A conceptual model has been formulated that considers multiple attributes at industry, project and individual levels that may at points in times become risk factors for workers poor mental health outcomes and subsequently could be addressed if there an awareness and understanding of these potentially modifiable levers.

We have presented a model to identify contextual factors in the occurrence of mental health issues in the construction industry via focusing on environmental elements across an entire project (from concept to completion). The model offers a mapping framework for the past research and the knowledge gaps that are yet to be addressed. As this is a conceptual model it is yet to be empirically evaluated. Future research could map stress points across the construction project and optimal points of intervention be it at a project management framework and/or individual support needs level. Modifiable factors that can be engaged to reduce stress throughout a construction project cycle have been identified within the model. A perceived limitation of our model is that it has not been yet tested. However, testing of the model is possible through evaluation of HR system data. Specifically, by measuring reactive changes one can evaluate work performance using HR data and individual employee wellness data. Future research could also lead to outcomes including recommendations and guidelines to engage management actors who can influence positive change through preventative strategies leading to effective and measurable mental health and project performance improvements. The research has practical implications for government policy makers, clients and developers, construction professional associations and businesses.

**Author Contributions:** Conceptualization, K.A.L. and T.M.; Literature Review, T.M. and K.A.L.; writing—original draft preparation, K.A.L., T.M. and C.M. writing—review and editing, K.A.L.; T.M. and C.M. visualization, K.A.L. and T.M. All authors have read and agreed to the published version of the manuscript.

**Funding:** This research received no external funding.

**Data Availability Statement:** Not applicable.

**Conflicts of Interest:** The authors declare no conflict of interest.

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
