# Peer review of "Healthier Construction: Conceptualising Transformation of Mental Health Outcomes through an Integrated Supply Chain Model"

_sustainability, doi:10.3390/su14159460_

Round 1

Reviewer 1 Report

  1. How can you identify contextual factors in the occurrence of mental health issues of worker?
  2. Please listed the restriction of the model
  3. Please added more recent citation. 
  4. Sustainable supply chain coordination with greening and promotional effort dependent demand.
  5. A three-layer supply chain inventory model for non-instantaneous deteriorating item with inflation and delay in payments in random fuzzy environment

Reviewer 2 Report

Discussion and conclusions should be strengthened. Explain clearly what the result of the research is and how it differs from previous research.

Theoretical and practical implications are weak.

Research limitations need to be fundamentally reconsidered, and most of them are incorrect.

please consider these papers in your manuscript:

Impact of coercive and non-coercive environmental supply chain sustainability drivers on supply chain performance: mediation role of monitoring and collaboration

A review on supply chain performance measurement systems

An Investigation on the Relationship between Supply Chain Agility and Financial Performance

Reviewer 3 Report

I have reviewed the following manuscript: Healthier Construction: Conceptualising Transformation of Mental Health Outcomes through an Integrated Supply Chain Model. The topic is very important, and thanks for the opportunity to review it.

This article present a transformational change model integrating construction industry specific context knowledge with psychosocial expertise to improve workers’ mental health.

This is an innovative and integrated model, with originality and novelty, which I believe will also interest to the readers.

I just have one question, I hope the author can clarify.

In Figure 1, the authors list the factors that affect workers' mental health, such as industry context, project performance, and project environment. But in this figure, I can't see where the workers are, so I can't understand what these factors are in relation to the workers, and how changing these factors affects the workers' mental health.

Author Response

Reviewer Comment

In Figure 1, the authors list the factors that affect workers' mental health, such as industry context, project performance, and project environment. But in this figure, I can't see where the workers are, so I can't understand what these factors are in relation to the workers, and how changing these factors affects the workers' mental health.

Authors Response

A very good point. The diagram has been changed to add in individuals. Additional text has also been added:

To respond to high rates of depression, anxiety and suicide in the construction industry, there is a need to generate new knowledge about the interactions between a) industry structure and behaviour b) project environmental characteristics and c) project performance metrics in relation to mental health outcomes and how the individual is embedded within project, organisational and industry supply chain context.

The conceptual table later in the paper explains in more detail the relationship between the factors and mental health outcomes. However it is the purpose of this paper to establish that this a gap and that this proposed conceptual model needs empirical evidence to identify where, how and to what extent and 'if' etc these factors influence mental health outcomes.